# In Vivo Quantitative Vasculature Segmentation and Assessment for Photodynamic Therapy Process Monitoring Using Photoacoustic Microscopy

**DOI:** 10.3390/s21051776

**Published:** 2021-03-04

**Authors:** Thi Thao Mai, Su Woong Yoo, Suhyun Park, Jin Young Kim, Kang-Ho Choi, Chulhong Kim, Seong Young Kwon, Jung-Joon Min, Changho Lee

**Affiliations:** 1Department of Artificial Intelligence Convergence, Chonnam National University, Gwangju 61186, Korea; 196286@jnu.ac.kr; 2Department of Nuclear Medicine, Chonnam National University Hwasun Hospital, Hwasun, Jeollanamdo 58128, Korea; yoosw.md@gmail.com (S.W.Y.); kwonsy@chonnam.ac.kr (S.Y.K.); jjmin@jnu.ac.kr (J.-J.M.); 3Interdisciplinary Program of Molecular Medicine, Chonnam National University, Gwangju 61186, Korea; suhyeonpark78@gmail.com; 4Department of Creative IT Engineering and Electrical Engineering, Pohang University of Science and Technology (POSTECH), 77 Cheongam-ro, Nam-gu, Pohang, Gyeongbuk-do 37673, Korea; ronsan@postech.ac.kr (J.Y.K.); chulhong@postech.edu (C.K.); 5Department of Neurology, Chonnam National University Hospital, 8 Hak-dong, Dong-gu, Gwangju 501-757, Korea; ckhchoikang@chonnam.ac.kr; 6Department of Nuclear Medicine, Chonnam National University Medical School, Jeollanamdo 58128, Korea

**Keywords:** photodynamic therapy, photoacoustic microscopy, quantitative analysis

## Abstract

Vascular damage is one of the therapeutic mechanisms of photodynamic therapy (PDT). In particular, short-term PDT treatments can effectively destroy malignant lesions while minimizing damage to nonmalignant tissue. In this study, we investigate the feasibility of label-free quantitative photoacoustic microscopy (PAM) for monitoring the vasculature changes under the effect of PDT in mouse ear melanoma tumors. In particular, quantitative vasculature evaluation was conducted based on Hessian filter segmentation. Three-dimensional morphological PAM and depth-resolved images before and after PDT treatment were acquired. In addition, five quantitative vasculature parameters, including the PA signal, vessel diameter, vessel density, perfused vessel density, and vessel complexity, were analyzed to evaluate the influence of PDT on four different areas: Two melanoma tumors, and control and normal vessel areas. The quantitative and qualitative results successfully demonstrated the potential of the proposed PAM-based quantitative approach to evaluate the effectiveness of the PDT method.

## 1. Introduction

Photodynamic therapy (PDT) has been widely used as an alternative therapeutic modality against cancer since its first oncologic application in 1972 [1]. PDT damages the target tissue through the interaction of three components—photosensitizer (PS), light at an appropriate wavelength, and oxygen dissolved in the cells [2,3]. Under proper wavelengths, the light-activated PS kills tumor cells. The advantage of PDT is its high selectivity to malignant lesions, while minimal damage in nonmalignant tissues reduces its toxicity [4]. The therapeutic mechanism of PDT can be described based on the combination of three categories: (1) Direct tumor cell damage by releasing cytotoxic agents; (2) stimulating inflammatory and immune responses of the body toward tumor cells; and (3) damage to the vasculature to enhance necrosis of the tumor cells [5]. Regarding vascular damage, PDT induces vasoconstriction, blood flow stasis, and thrombus formation in a targeted vessel [6]. The blocked supply of oxygen and nutrition to the tumor causes necrosis. To evaluate the therapeutic efficacy of PDT on tumors, quantitative methods that can measure vascular responses are needed.

Various imaging modalities have been applied to monitor the therapeutic response using PDT, including computed tomography (CT), positron emission tomography (PET), magnetic resonance (MR) imaging, and optical or ultrasound (US) imaging [7]. However, each modality has limitations. Both CT and PET imaging require radiation exposure, and patients are susceptible to its associated risks. MR has a toxicity problem because of the necessary use of contrast agents [8]. Additionally, these imaging methods have low resolution, which makes it challenging to visualize micro-vessels, and they require a post-processing process to provide reliable images. Thus, these clinical imaging modalities are not suitable for monitoring the therapeutic responses of micro-vessels. US imaging provides cross-sectional structural and functional images in real-time but has relatively low resolution and sensitivity. Optical imaging enables the obtaining of a high-resolution image at high speed, but the shallow imaging depth is an issue [9]. Therefore, a safe, alternative label-free, vasculature imaging modality with higher imaging resolution is required.

The photoacoustic (PA) effect involves the conversion of light into sound waves. When a photon beam illuminates tissue, some photons are absorbed by molecules, and their energy is partly converted to heat, causing thermal expansion. This thermal expansion generates sound waves that propagate through the tissue. Based on the PA effect, photoacoustic imaging (PAI) provides hybrid (optical and US) imaging characteristics [10,11,12]. Unlike conventional US imaging, in which the contrast depends on the tissue’s elasticity and mechanical properties, PAI inherits the contrast provided by the absorption of light from the optical property. Thus, selecting the proper light wavelength enables the detection of chromophores inside tissues such as hemoglobin, lipids, tendons, and melanin [13,14,15,16,17]. Compared to typical optical imaging modalities, PAI achieves a relatively high penetration depth owing to less ultrasound scattering. With PAI, anatomical structures are easily visualized, and functional information such as blood oxygenation, oxygen saturation, blood flow, and metabolism can be obtained [18,19]. Owing to its flexible imaging system configuration, which allows a specific resolution and penetration depth, PAI is used in various biomedical applications [20,21,22,23,24,25,26,27,28,29]. PA computed tomography (PACT) focuses on reconstructing an image from one-dimensional radial information defining the relationship between the PA source and the detector position. PACT has flexibility because it uses multi-arrayed transducers to achieve a high penetration depth [30,31,32,33,34,35]. By contrast, PA microscopy (PAM) does not require reconstruction algorithms to generate three-dimensional images. Using a focused or unfocused ultrasonic transducer to detect the PA signal, PAM can directly obtain a one-dimensional image for each detection, which combined with a 2D raster scan, yields high-resolution 3D images. Furthermore, based on its superior resolution and sensitivity, PAM is widely used for vascular network imaging and for monitoring vasculature activity [36,37,38,39,40,41,42].

PAI has significantly contributed to monitoring and evaluating the effectiveness of PDT as a guided imaging method [43,44,45]. In particular, studies using PAM can evaluate the impact of PDT at the micro-vessel level thanks to its higher resolution. Shao et al. [46] used PAM to investigate vascular changes following PDT treatment. However, in addition to the PAM images, only vessel diameter and oxygen saturation were monitored. To further assess the effectiveness of PDT, Rohrbach et al. [47] combined PAM and diffuse correlation spectroscopy to extract vessel area, vessel diameter, and blood flow in nonmelanoma skin cancer. However, the diameter and vessel area are not sufficient by themselves to reflect the variability in vasculatures, and there are many other specific vascular parameters that should be monitored. To achieve an intensive micro-vessel investigation, a method that applies vasculature segmentation such as the Hessian filter [48] to obtain advanced quantitative values is required.

This study monitored the changes in melanoma tumors and surrounding microvascular structures under the PDT process via a lab-built PAM system. First, rose bengal (RB) was used as the PS. After injecting it into a mouse with two melanoma tumors on its ear via intravenous injection, the PDT process was conducted twice (one treatment per tumor) by applying light-emitting diode (LED) laser illumination for 20 s each time. For each step of the experiment, an optical-resolution PAM (OR-PAM) system was used to monitor changes in vascular structure. Based on the acquired three-dimensional OR-PAM data, we reconstructed the maximum amplitude projection (MAP) of the PAM images and applied a quantitative evaluation process using adaptive Hessian filter segmentation. Finally, a qualitative analysis using MAP and depth-reserved OR-PAM images and a quantitative analysis based on extracting five specific vascular parameters consisting of the PA signal, diameter, density, perfused vessel density (PVD), and complexity were performed.

## 2. Materials and Methods

### 2.1. Experimental Setup for OR-PAM

Figure 1a shows the experimental setup of the OR-PAM system for PDT process monitoring. In the OR-PAM imaging section, a nanosecond pulsed laser (SPOT-10-200-532, Elforlight, 532 nm central wavelength, 6 ns pulse width, 10 kHz repetition rate) served as the light source for OR-PAM. The pulsed laser beam was delivered to a collimator (F280APC-A, Thorlabs), which provided easy coupling with a single-mode optical fiber (P1-405BPM-FC-1, Thorlabs) with over 60% coupling efficiency. The laser beam reached the second collimator of the OR-PAM probe via the optical fiber and became a collimated beam with a 2 mm diameter. This collimated beam was then focused by a doublet lens (AC254-060-A, Thorlabs) and passed through a homemade beam combiner. The beam combiner is composed of normal and aluminum-coated prisms to reflect the laser beam perpendicularly and transmit the captured acoustic wave, which provides a co-confocal alignment condition between the laser beam and the generated PA wave. An acoustic lens (NT45-010, Edmund) was attached to the front of the beam combiner to generate the focused acoustic capturing regime. Then, using a hybrid scanning system that combines a one-axis-MEMS scanner (OpitchoMS-001, Opticho Inc., Ltd.) for fast B-scanning, a linear scanning motor (PT1-Z8, Thorlabs) for wide mosaic scanning on the X-axis, and a linear scanning motor (L-509-10SD00, PI) on the Y-axis, the focused laser beam setup provided the desired volumetric mosaic scanning function. After the focused laser beam was projected onto the sample, the resulting PA wave propagated to the beam combiner and was detected by a high-frequency transducer (V214-BC-RM, 50 MHz, Olympus) under the intensive acoustic capture condition. The acquired PA signal was then amplified by two serially connected RF-amplifiers (ZX60-3018G-S+, Mini-Circuit) and digitalized using a high-speed digitizer (ATS9371, AlazarTech) with 12-bit resolution and 1 GS/s sampling rate. A data acquisition board (DAQ) (PCIe-6321, NI Instruments) controlled the hybrid scanning system. The laser was controlled at approximately 5 mJ/cm^2^ below the maximum ANSI limit (20 mJ/cm^2^ under visible light). The measured axial and lateral resolutions were 27 and 14 µm, respectively [49], and are well matched to the theoretical values. The single B-scan was displayed at 1064 × 200 pixels using a 25 Hz frame rate. All data analyses and reconstructions were performed using MATLAB (R2017b, Mathworks).

### 2.2. Experimental Setup for PDT

RB (95% certified dye content, 330000-5G, Sigma-Aldrich, Inc.) was dissolved into phosphate-buffered saline (PBS) to obtain a 10 mg/mL concentration solution and was then sterilized by passage through a 0.22 µm filter. The RB solution (100 µL) was injected into the nude mouse intravenously through its tail. To induce the photodynamic effect, a green LED at 525 nm (TouchBright X-6, Live cell instrument) irradiated the mouse ear, as shown in Figure 1b. The effective LED laser beam illuminated an elliptically shaped area (3.9 mm × 5.1 mm) at 150 mW/cm^2^. We conducted the LED laser illumination for 20 s twice.

### 2.3. Melanoma Cell Culture

B16F10 murine melanoma cells were used for the ear tumor model. The cell line was cultured in Dulbecco’s modified Eagle’s medium (DMEM) with 4500 mg/L D-glucose, L-glutamine, 110 mg/L sodium pyruvate, 10% fetal bovine serum (FBS), and 1% penicillin-streptomycin. The medium was replaced every 3–4 days. The cells were incubated at 37 °C in a humidified atmosphere with 5% CO_2_. All media and reagents were purchased from Welgene Inc. (Gyeongsangbuk-do, Korea).

### 2.4. Animal Preparation and Melanoma Tumor Bearing

A BALB/c nude mouse (8 week-old female, Orientbio) was prepared for in vivo experiments. All animal experiments were carried out in accordance with the guidelines of Chonnam National University Hwasun Hospital. The mouse was anesthetized with a mixture of ketamine (87.5 mg/kg) and xylazine (12.5 mg/kg). B16F10 murine melanoma cells were harvested, pelleted by centrifugation at 300 g for 5 min, and suspended in PBS. The obtained melanoma cell suspension consisted of 1 × 10^5^ melanoma cells in 10 µL of PBS and was implanted subcutaneously into the right ear. Melanoma cells were injected into two regions, as shown in Figure 1a.

### 2.5. Protocol of PAM-Monitored Melanoma PDT Treatment Experiment

The process of the experiment is described in Figure 2. The PAM system was used for imaging before and after each stage of the experiment including RB injection, 1st PDT, and 2nd PDT. With a mouse ear size of 10 mm × 12 mm, it requires 3 min for 3D image data acquisition time. The time to implement RB injection is ~1 min. The duration of each PDT treatment is 20 s. Therefore, the total time for conducting this experiment is approximately 14 min.

### 2.6. Quantitative PAM Image Analysis Method Based on Hessian Filter Segmentation

Figure 3a shows the flowchart for the comprehensive quantitative evaluation, which includes morphological evaluation, assessment based on intensity values, and blood vessel parameters. First, the mouse ear was scanned by the OR-PAM system to obtain the three-dimensional volumetric OR-PAM data. Based on the Hilbert transform and using only the maximum amplitude information, a 2D reconstructed image, which is called the MAP image, was obtained (Figure 3b(i)). A median filter method was adopted to partially remove the noise in the MAP image to improve the morphological evaluation. For image intensity assessment, we calculated the mean value of the intensity in the areas of interest. Due to the significant standard deviation variation, the data were arranged in an order from low to high, and the upper 50% of the data were used.
(1)$$PA\;signal=\frac{\sum_{i=1}^{m}\sum_{j=1}^{n}I\left(i,j\right)}{m*n}.$$
In the above equation, I(i,j) is the intensity at a point (i,j) of the MAP image and *I* should be higher than 50% of the mean value of the entire MAP’s intensity; [m,n] is the size of the image. In this section, m and n define the size of all images discussed. The *PA* signal is the main factor affecting the displayed image. In the same material type, a higher *PA* signal value indicates a higher density of that material. For the quantitative evaluation, we aimed to extract four primary parameters of blood vessels: Blood vessel diameter (*VDI*), blood vessel density (*VD*), *PVD*, and blood vessel complexity (*VC*). As a mandatory requirement, the OR-PAM MAP image was segmented. Here, we used a multi-scale Hessian filter to enhance any multi-size blood vessel structures. The result of the Hessian filter process is shown in Figure 3b(ii). Then, an adaptive threshold was applied to acquire a binary image (Figure 3b(iii)). The yellow pixels have the value of 1, and the remaining pixels belong to the background with a value of 0. *VD* is calculated as the ratio represents the number of yellow pixels per total pixels of the binary image.
(2)$$VD=\frac{\sum_{i=1}^{m}\sum_{j=1}^{n}Y\left(i,j\right)}{m*n},$$
where Y(i,j) represents a yellow pixel on the binary image. *VD* shows the area occupied by the blood vessels. In the case of a tumor, an increase in *VD* indicates strong tumor growth. The skeleton image in Figure 3b(iv) was defined as an image showing only the centerline and was obtained by morphological operation implementation on the binary image. *PVD* is determined by computing the ratio of red skeleton pixels per total skeleton pixels in Figure 3b(iv).
(3)$$PVD=\frac{\sum_{i=1}^{m}\sum_{j=1}^{n}R\left(i,j\right)}{m*n},$$
where R(i,j) represents a red pixel on the skeleton image. As a representation of perfusion, *PVD* only considers a blood vessel’s existence, not the area in which that blood vessel is located. Thus, the changes in vascular perfusion inside the tumor can be reflected by *PVD*.

To calculate the *VDI*, we follow two principles. First, the shortest distance between the two edges of a vessel is considered to be its diameter. Second, the mean value of the diameter of a vessel is the average value along the centerline of that vessel. Therefore, we applied the Euclidean transform to the binary image along the skeleton line of the skeleton image. Briefly, *VDI* can be calculated as follows:(4)$$VDI=\frac{\sum_{i=1}^{m}\sum_{j=1}^{n}E\left(i,j\right)}{m*n},$$
where E(i,j) is the Euclidean distance transform. The variations in the size of any blood vessels are reflected in the *VDI*. For *VC* computation, the box-counting method was used [50].
(5)$$VC=\frac{\log(N_{s})}{\log(\frac{1}{s})}.$$
In the equation, s indicates the size of the unit box and $N_{s}\;$ is the number of boxes. *VC* represents the complexity of the vascular network.

We constructed the distributions of the quantitative parameters across the entire image to provide a visual overview. For each parameter, a window size of 8 × 8 pixels was used. The value of each parameter was calculated for this window area and then stored at the center point. By moving the window across the entire image, a quantitative map was created. Figure 3c(i–iv) display the quantitative maps consisting of the diameter map, density map, PVD map, and complexity map, respectively. The melanoma area and other areas such as the normal vasculature area and the area near the melanoma were monitored. Thus, we selected four small rectangular regions of interest (ROIs) to represent these areas. The effect of PDT on these ROIs is discussed in the Results section.

## 3. Results

### 3.1. In Vivo OR-PAM Monitoring of Melanoma Tumors during PDT

All OR-PAM imaging results and photographs after PDT treatment are shown in Figure 4. The first row (Figure 4a(i–iii)) shows control photographs of the mouse ear and the results after two PDT treatments. In particular, Figure 4a(ii) shows the irradiated area on the mouse ear. We attempted to irradiate the centers of the melanoma tumors with the PDT laser beam. The second row shows the OR-PAM MAP images (Figure 4b(i–iv)), which were acquired before PDT (Figure 4b(i)), after injection of the RB (Figure 4b(ii)), after the first PDT (Figure 4b(iii)), and after the second PDT (Figure 4b(iv)). The OR-PAM MAP image (Figure 4b(i)) clearly shows the entire structures of the two melanoma tumors and related blood micro-vessels. In particular, compared with the control photograph (Figure 4a(i)), the OR-PAM image revealed the correct boundaries of both melanoma tumors and related micro-vessels. After the RB injection (Figure 4b(ii)), a significant increase in the PA signal was observed in both the melanoma and vascular areas. Moreover, many small micro-vessels were visible as well. The PDT treatment process was divided into two stages to both remove the melanoma tumors and minimize the damage in unrelated areas. For the first stage, a 525 nm-wavelength PDT laser beam was used to irradiate the left melanoma tumor for 20 s, and the OR-PAM MAP image is shown in Figure 4b(iii). The blood vessels on the upper edge of the OR-PAM image, far from the melanoma, appeared to display less damage. By contrast, the melanoma and corresponding micro-vessels were damaged, and the blood vessels directly connected to the tumor were partially removed. After the second PDT treatment (Figure 4b(iv)), both melanomas were destroyed. Particularly, the right melanoma and its related micro-vessels were eliminated, while surrounding normal vessel areas remained undamaged.

The third row shows the depth-resolved OR-PAM MAP images (Figure 4c(i–iv)), corresponding to the second-row OR-PAM MAP images. Before PDT treatment (Figure 4c(i–ii)), the two tumors were most prominent, with a maximum elevation of approximately 400 µm, as measured from the deepest blood vessel. As shown in Figure 4c(iii), after the first PDT, most of the vessels around melanoma 1 disappeared, and the height of melanoma 1 was significantly reduced. By contrast, the first PDT had less effect on melanoma 2 and its vessels. Figure 4c(iv) indicates the depth of the tumors after the second PDT, indicating that melanoma 2 was completely destroyed. The trace remaining at the location of melanoma 1 in Figure 4b(iv) is the deepest signal part, indicating that melanoma 1 was almost completely destroyed.

Figure 4d–g display the contribution maps of the four quantitative parameters: Diameter, density, PVD, and complexity. All maps indicate severe signal deterioration in the melanoma 1 area and the complete disappearance of melanoma 2 after two PDT treatments. In particular, in the density map and PVD map, before applying the PDT process, the melanoma areas display the highest signal concentrations (Figure 4e(i–ii),f(i–ii)). After the first treatment, the signal concentration was detected primarily in melanoma 2 (Figure 4e(iii),f(iii)). After the second PDT, as shown in Figure 4e(iv),f(iv), only signals in areas far from the tumors were observed, while signals associated with melanoma 1 and melanoma 2 disappeared.

For detailed observation of the effects of PDT in the micro-vessels, we extracted and monitored four small ROIs. ROI A is the control vessel area. ROI B is the normal area near the area of melanoma 1, and ROIs C and D represent melanoma 1 and melanoma 2, respectively. Figure 5 shows images of these ROIs before and after PDT treatment. As with the whole image, RB injection induced an increase in all parameters at every ROI. The rise in PA signal was especially prominent in ROI C. About PDT treatment compacts, each ROI should be independently assessed. The least impact was observed at ROI A. After the first PDT, the stability of all quantitative values was shown in all quantitative maps. The second PDT caused harm at the large vessel of ROI A that was visualized at the PA signal map and diameter map, and also caused the disappearance of some vasculature in the small vessel area that was displayed in maps of density, PVD, and complexity. However, these effects were insignificant compared to other remaining ROIs. Melanoma 2 (ROI D) got the most serious impact from the PDT treatment process, especially the second PDT. The first PDT mainly instigated the breaking of the shape for the large vessel directly connected to melanoma 2 on the diameter map. After the second PDT, all quantitative maps displayed only some tiny spots remaining, and the signals in the melanoma 2 area disappeared, indicating the complete obliteration of melanoma 2. Melanoma 1 (ROI C) had the greater height compared to melanoma 2 that was also destroyed. The first PDT broke the structure of melanoma 1 and its large nearby vessel. Thus, the unshaped area has been shown on the diameter map instead of melanoma and vessel morphology. Although the center area still appeared on the remaining maps, the edge area of melanoma 1 was almost injured. After the second PDT, melanoma 1 was not completely damaged but the change in the center of melanoma 1 on all quantitative maps implies harm induced by the second PDT. ROI B is a normal vessel area next to melanoma 1 that got damaged as well. At the right area of ROI B nearest melanoma 1, quantitative parameters got a value of almost 0 after the PDT process. Its signal almost disappeared, and we could not observe the vascular morphology in the after photograph and other quantitative maps.

### 3.2. Quantitative Assessment of Microvasculature Changes Using Photodynamic Therapy

Four ROIs extracted in the previous section were continuously monitored using the quantitative process described in Section 2.6. To normalize the units for all parameters, the value before the RB injection was chosen as the control value. Other values were calculated based on the difference with the control value and are presented as a percentage. To clearly observe the effect of PDT treatments on different regions, a comparison was conducted among all four ROIs for each of the five parameters: (1) PA signal, (2) VDI, (3) VD, (4) PVD, and (5) VC. All results are summarized in Figure 6. In general, the trend of all vessel parameters is to increase after RB injection, and then exhibit a sharp decrease after the first PDT treatment followed by a lesser decrease after the second PDT. The first parameter, the PA signal, is shown in Figure 6a. Only ROI A exhibited a PA signal approximating the control value after two PDT treatments. For other ROIs, the PA signal slowly increased after the RB injection with 5%, 7.8%, and 6.5% increases for ROIs B, C, and D, respectively. Then, the first and second PDT treatments caused the signal to drop suddenly and continue to the decrease in these ROIs. In particular, ROIs B and C were strongly influenced by the first PDT, causing their values to decrease by 67% and 48.5%, respectively. After the second PDT, this value only decreased by 5% compared to the first PDT, on the ROI C region. By contrast, the first PDT slightly changed the PA signal value of ROI D (its value remained similar to the control value), but the second PDT caused it to decrease by 62%. Among the vessel-specific parameters including VDI (Figure 6b), VD (Figure 6c), PVD (Figure 6d), and VC (Figure 6e), the least variation was found for the VC, in which the mean value of all ROIs was 4.1% after RB injection, −5.5% for the first PDT, and −12.2% for the second PDT. In addition, the largest change occurred in the VD: 13.4%, −18.7%, and −42.1% after the RB injection, the first PDT, and the second PDT, respectively. The minus symbol implies a decrease in value. Regarding the effect of PDT on the ROIs, the maximum change in VDI was only −12.55% for ROI A after the second PDT treatment and was found to be the most stable area. By contrast, ROI C (covering melanoma 1) exhibited the largest decreases among the ROIs after the first PDT: −29% for VDI, −39.8% for VD, −21.4% for PVD, and −16% for VC. The corresponding values after the second PDT were −36%, −60.4%, −41.6%, and −27%. After two PDT treatments, for melanoma 2 (ROI D), the VDI, VD, PVD, and VC values were −14.2%, −47.2%, −42%, and −17.8%, respectively. Unfortunately, a normal blood vessel near melanoma 1 (ROI B) was also damaged, with a 43% decrease in density and 26% decrease in PVD.

## 4. Discussion

We reported the vasculature response after short-term PDT treatments using both qualitative and quantitative success PAM monitoring for areas within and surrounding two melanoma tumors. In particular, using the microscale-resolution OR-PAM imaging and precise segmentation approach, changes in microvasculature structure near melanoma tumors during PDT treatment was successfully observed without any ultrasound imaging guiding. Qualitatively, the disappearance of the two tumors and their related micro-vessels were visualized on OR-PAM MAP and depth-resolved images using small ROIs. Quantitatively, we obtained five vascular parameters: The PA signal, VDI, VD, PVD, and VC. By monitoring and comparing these parameters before and after the PDT process, we can arrive at some conclusions. First, the two melanomas were almost completely destroyed. In the case of melanoma 1, 53.8% of the PA signal, 60.4% of the VD, and 41.6% of the PVD were removed after conducting two PDT treatments. The corresponding values for melanoma 2 were 62%, 47.2%, and 42%. In addition, VDI and VC exhibited a decrease in nearly 30% in both melanomas. Second, PDT only has a significant impact on the target area. This is evidenced by a strong drop in the value of the PA signal (−48.5%) and VDI (−29%) in ROI C after the first PDT, which targeted ROI C, and only dropped another 5% after the second PDT, which targeted ROI D. Third, not only the melanoma area but also the normal area was monitored. Unfortunately, a normal blood vessel near melanoma 1 (ROI B) was also damaged, with a 43% decrease in density and 26% decrease in PVD. However, these values are lower than the values in the melanoma areas targeted in this study [47].

Although the proposed method was successful in evaluating the vascular response induced by PDT, there are some limitations. One of the main reasons that prevent OR-PAM from becoming a clinical imaging tool is the shallow penetration depth (~1 mm), which limits the size of the melanoma that can be evaluated using the proposed method [51,52]. To overcome this issue, an AR-PAM system with higher penetration depth at the same laser wavelength can be a good alternative [53]. In particular, the integrated system OR-AR-PAM [54,55] could solve the depth issue in the melanoma area while maintaining the high resolution for monitoring the surrounding microvessels. Second, the 2D Hessian filter is limited when distinguishing between a tubular shape and plate-like shape and can erroneously classify the melanoma region center as several blood vessels that share a common origin. Hence, it is necessary to either apply a 3D quantitative process with a 3D Hessian filter or to separate the vessel and melanoma area when filtering for more accurate analyses [48]. Third, changes in oxygen saturation induced by PDT treatment were demonstrated [46]. Thus, for a comprehensive assessment of PDT, the quantitative process presented in this report should be performed concurrently with the monitoring of the functional information of blood vessels, such as SO_2_ concentration and blood flow [19]. However, as a limitation of using a single-wavelength laser, extracting SO_2_ concentration information is not possible for our current system. Using two more different suitable laser wavelengths is a solution to this issue [56]. Finally, in this study, only one mouse was used and presented the result as a pilot study. For further assessment of the PDT effectiveness, it would be better to implement this approach on many mice, especially with different sizes of melanoma tumors. The high spatial resolution, portability, and fast imaging of the PAM system are the key factors for accelerating clinical translation. Hence, laser diodes [57] or LEDs [58] assuring enough energy, compact-size, and high repetition rates can be the alternatives for current expensive and massive laser sources. This approach is cost-effective and a worthy direction to consider toward a clinical transition.

## 5. Conclusions

We successfully implemented the proposed OR-PAM technique for monitoring changes in melanoma tumors and related micro-vessels under the effects of the PDT process. Moreover, three-dimensional OR-PAM data were gathered to acquire high-resolution MAP and depth-resolved OR-PAM images for qualitative analysis, and to obtain tumor microvascular information, including the PA signal and VDI, VD, PVD, and VC, for quantitative analysis. Based on the results, we believe that the proposed quantitative PA evaluation approach can contribute to precise tumor therapy monitoring.

## Figures and Tables

**Figure 1 sensors-21-01776-f001:**
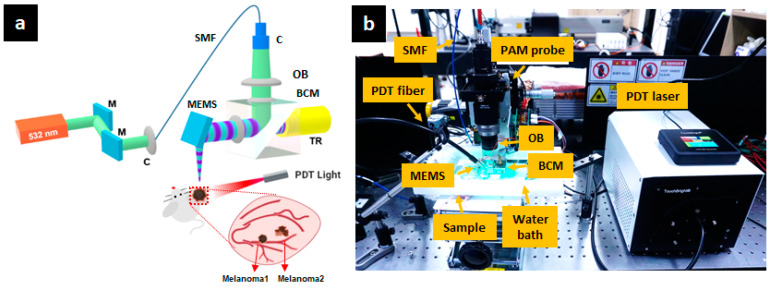
Photodynamic therapy (PDT) monitoring of melanoma tumors in a mouse ear using optical-resolution photoacoustic microscopy (OR-PAM). (**a**) Experimental setup of OR-PAM for monitoring PDT process, (**b**) photograph of the experimental setup. M, mirror; C, collimator; OB, objective lens; BCM, beam combiner; SMF, single-mode fiber; TR, transducer.

**Figure 2 sensors-21-01776-f002:**
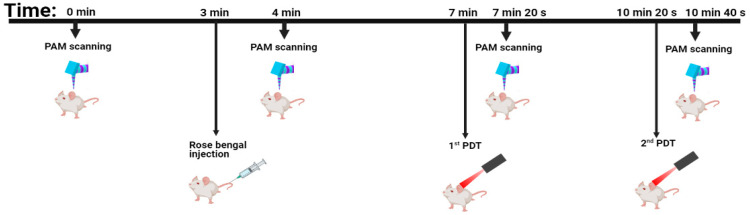
Protocol of the PAM-monitored melanoma PDT treatment.

**Figure 3 sensors-21-01776-f003:**
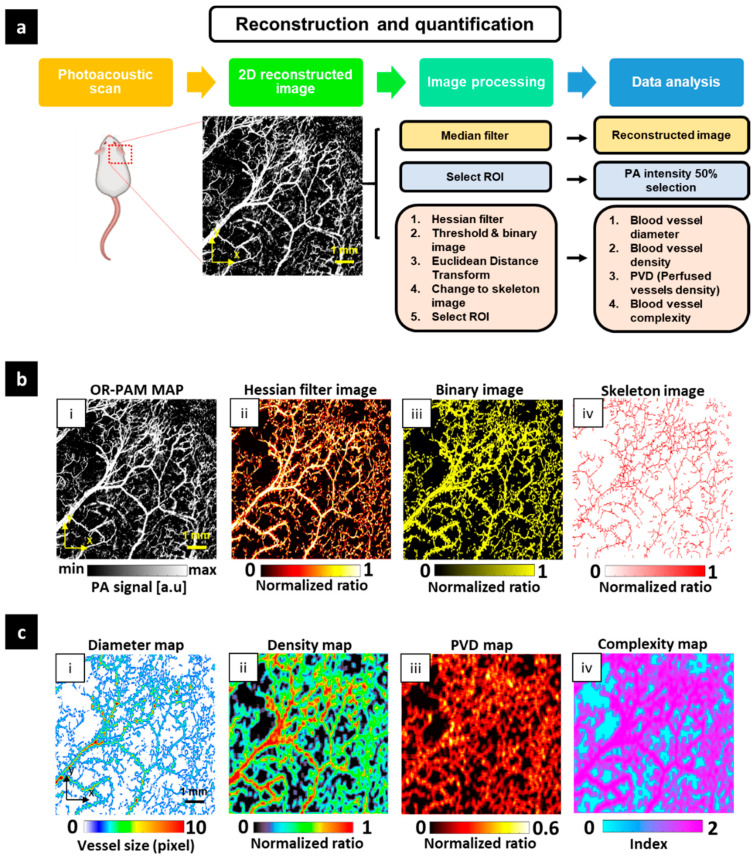
Flow chart of quantitative PAM image analysis based on Hessian filter segmentation. (**a**) Quantitative process flow chart. (**b**) Image segmentation and processing steps; b(i) OR-PAM maximum amplitude projection (MAP), b(ii) Hessian filter image, b(iii) binary image, b(iv) skeleton image. (**c**) Quantitative parameter maps; c(i) diameter map, c(ii) density map, c(iii) perfused vessel density (PVD) map, and c(iv) complexity map.

**Figure 4 sensors-21-01776-f004:**
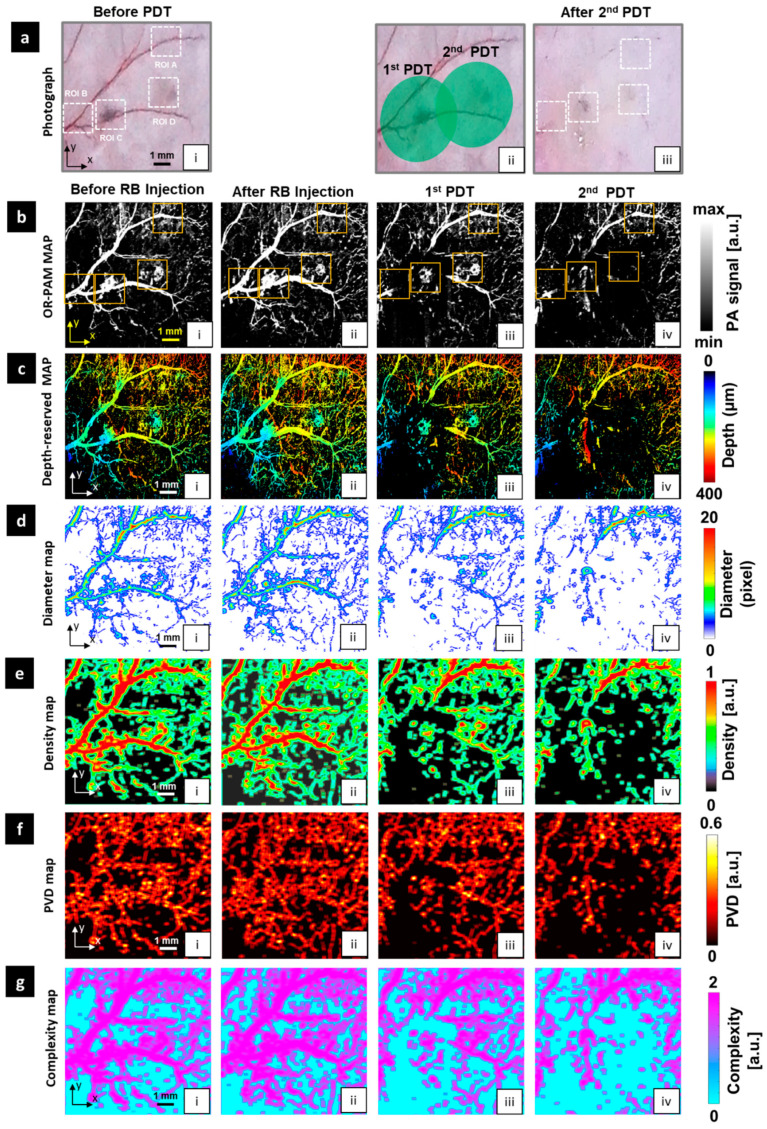
In vivo OR-PAM monitoring of melanoma tumors and micro-vessels in the mouse ear before and after two PDT applications. (**a**) Photographs of mouse ear; a(i) before PDT, a(ii) after PDT beam irradiation, a(iii) after two PDT treatments. (**b**) OR-PAM MAP images. (**c**) Depth-resolved OR-PAM MAP images. (**d**) Diameter map. (**e**) Density map. (**f**) PVD map. (**g**) Complexity map. For all images for (**b**–**g**): (i) Before rose bengal (RB) injection, (ii) after RB injection, (iii) after 1st PDT, and (iv) after 2nd PDT.

**Figure 5 sensors-21-01776-f005:**
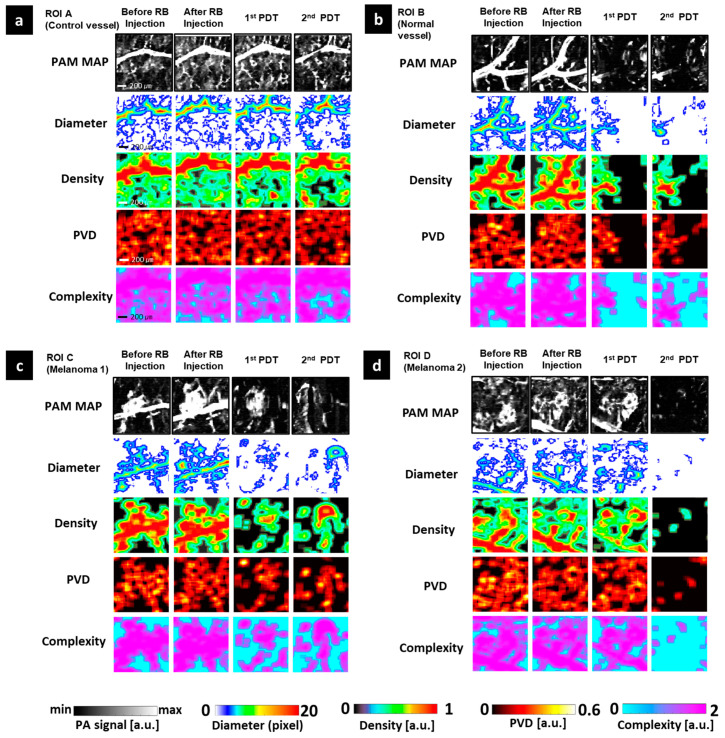
OR-PAM image and quantitative maps of four selected regions of interest (ROIs). (**a**) ROI A (control vessel). (**b**) ROI B (normal vessel). (**c**) ROI C (Melanoma 1). (**d**) ROI D (Melanoma 2). ROI, region of interest.

**Figure 6 sensors-21-01776-f006:**
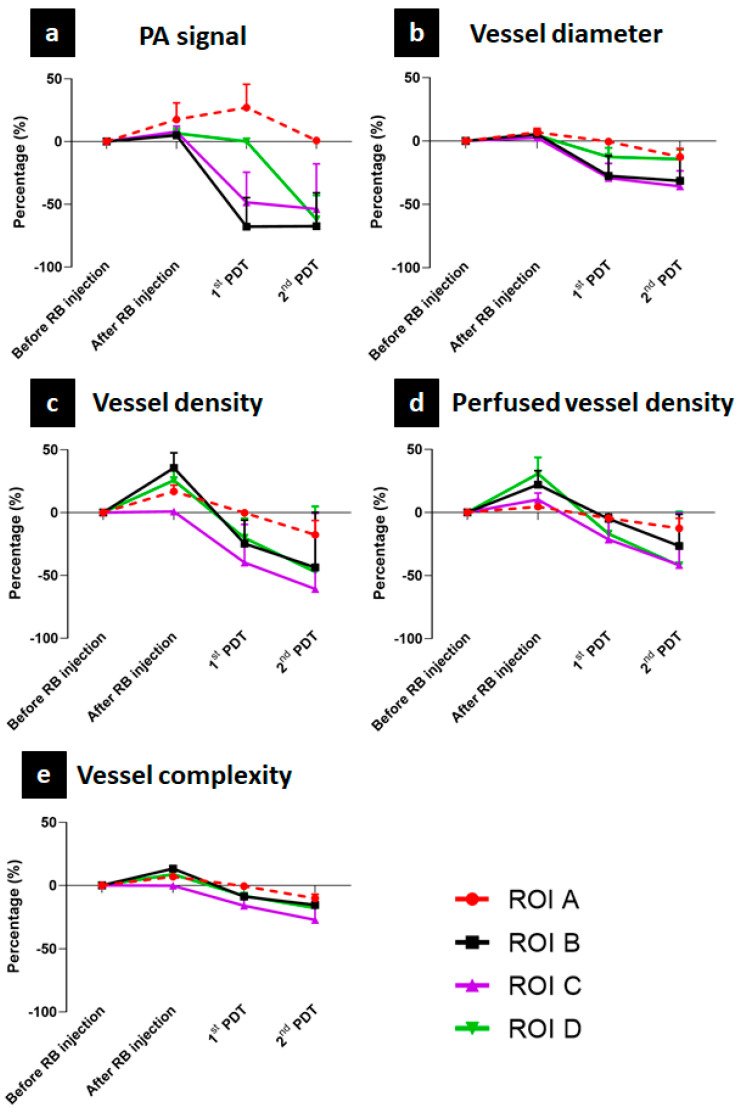
Quantitative evaluation results among ROI A (control vessel), ROI B (normal vessel), ROI C (melanoma 1), and ROI D (melanoma 2). (**a**) PA signal, (**b**) vessel diameter, (**c**) vessel density, (**d**) perfused vessel density, (**e**) vessel complexity.

## Data Availability

Not applicable.

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
