# Peer review of "In Vivo Quantitative Vasculature Segmentation and Assessment for Photodynamic Therapy Process Monitoring Using Photoacoustic Microscopy"

_sensors, 2021, doi:10.3390/s21051776_

Round 1

Reviewer 1 Report

The manuscript presents quantitative vascular segmentation and assessment using PAM images for studying PDT process. Melanoma tumor models were used, and 2-times PDT were applied. 4 regions were monitored with 5 parameters: PA signal, VDI, VD, PVD, and VC. The results were presented qualitatively and quantitatively. The work is technically sound, and the manuscript is clearly written. There are some comments from the reviewer.

  1. Since this is just a study from one mouse, authors are suggested to discuss whether it would be helpful to conduct more mice in future.
  2. It will be helpful to provide a timeline (or a rough timeline), maybe using a figure, about when the PDT and PAM were conducted. For example, RB injection (at the 0 mins) -> PAM imaging (at the 1 mins) -> 1st PDT application (at the 5 mins) -> PAM imaging (at the 6 mins) -> 2nd PDT application (at the 10 mins) -> PAM imaging (at the 11 mins).
  3. In page 2: “This study monitored melanoma tumors and closed microvascular changes under the PDT process via a lab-built PAM system.” It is not clear about “closed microvascular changes” here. Please consider rephrasing.
  4. In page 3, line 103: “twice times” times is redundant.
  5. In page 5, line195-196: “The white pixel ratio between the total pixels of the binary image and the skeleton image is represented for VD or PVD as follows.” It is confusing here. First, what does it mean by white pixels? No “white” pixels are observed in Fig. 2b(iii). Also, I suppose VD is calculated from Fig. 2b(iii), while PVD is calculated from Fig. 2b(iv). If so, authors may consider rephrasing the sentence because it seems to say that a ratio is calculated using BOTH Fig. 2b(iii) and Fig. 2b(iv).
  6. In page 8, line 280-282: “Same with whole image, RB injection induced the increase at all ROIs, especially ROI C, which is visible for all quantitative maps.” It is not clear about what is increased, e.g., PA signals or other parameters?
  7. In Fig. 3a(ii), PDT regions are indicated. Please elaborate on the beam profile, e.g., a Gaussian beam profile? If so, does the boundary of the green circle indicate the -3 dB light intensity?
  8. In page 9, line 323-324, “After the second PDT, these values only decreased by 5% compared to the first PDT” Please indicate that this refers to ROI C.

Author Response

Thank you for your fruitful review. 

Please, see the attached our response.

Reviewer 2 Report

In this paper (In vivo Quantitative Vasculature Segmentation and Assessment for Photodynamic Therapy Process Monitoring using Photoacoustic Microscopy), Thi Thao Mai et al  investigated the feasibility of photoacoustic microscopy in monitoring the vasculature changes caused by PDT. Five quantitative vasculature parameters, including the PA signal, vessel diameter, vessel density, perfused vessel density, and vessel complexity were analyzed to evaluate the influence of PDT in a preclinical setting. Manuscript is interesting and well written and can be accepted for publication after following comments are addressed.

  • Use of photoacoustic imaging in guiding PDT treatment is a well explored topic. Authors must clarify the novelty in this work, especially focusing on clinical relevance if any. With limited PAM imaging depth (as mentioned in the discussion), how and why this approach is useful in melanoma imaging application?
  • Authors must discuss about Photoacoustic monitoring of PDT‐induced oxygen saturation changes and its importance in detecting therapeutic efficiency. Since oxygenation imaging is possible with PAM too, why this was not included in the study? Please discuss!
  • I assume that structural and morphological changes of a tumor tissue can be evaluated using clinical ultrasound, which is a well-accepted imaging modality. In a photoacoustic tomography setup, it may be straightforward to perform dual mode PA and US imaging since acoustic detection can be shared. This may be helpful to get structural and functional details in a single measurement. Please explain why this method will be more useful. Maybe it is important to discuss whether only enhanced resolution will help in this application!
  • If spatial resolution is important for this application, will it be possible to replace the expensive laser source with a laser diodes (https://spie.org/Publications/Book/2566565?SSO=1) or LEDs (https://link.springer.com/book/10.1007/978-981-15-3984-8) to perform acoustic resolution PAM imaging ? This may be an interesting development for accelerating clinical translation according to me. Please discuss!

Author Response

Thank you for your fruitful comments. 

Reviewer 3 Report

  1. PAT branches into 2 imaging modalities PACT and PAM. PAM is a high-resolution imaging branch of PAT. Please make these corrections in lines 78-83.
  2. In PAM, sometimes unfocused transducer is also used. Please rephrase line 83.
  3. Please define the total time taken to acquire one 3D image data.
  4. All figures does not have any area scale.
  5. How is the penetration depth and axial and lateral resolution calibrated for the system?
  6. Statistical analysis should be performed on all the quantitative data presented in Fig. 5.
  7. Along with the limitation in the research, alternative solutions should also be discussed.
  8. To overcome shallower penetration depth, ARPAM can be a good alternative with higher resolution as compare to PACT. Please comment and add this to the paper.
  9. Please add the following lietaryre to yoru introduction: Three-dimensional Hessian matrix-based quantitative vascular imaging of rat iris with optical resolution photoacoustic microscopy in vivo, by H Zhao. Subwavelength-resolution label-free photoacoustic microscopy of optical absorption in vivo, by C Zhang. Noninvasive photoacoustic computed tomography of mouse brain metabolism in vivo, by J Yao. Wide-field two-dimensional multifocal optical-resolution photoacoustic-computed microscopy, by J Xia. Optical-resolution photoacoustic microscopy for in vivo imaging of single capillaries, by K Maslov.

Author Response

Thank you for your fruitful comments. 

Round 2

Reviewer 2 Report

Authors addressed all my concerns. I recommend to accept this paper in present form.